# Overexpression of the *Aspergillus fumigatus* Small GTPase, RsrA, Promotes Polarity Establishment during Germination

**DOI:** 10.3390/jof6040285

**Published:** 2020-11-13

**Authors:** Adela Martin-Vicente, Ana C. O. Souza, Ashley V. Nywening, Wenbo Ge, Jarrod R. Fortwendel

**Affiliations:** Department of Clinical Pharmacy and Translational Science, University of Tennessee Health Science Center, Memphis, TN 38163, USA; amartinv@uthsc.edu (A.M.-V.); aolivei5@uthsc.edu (A.C.O.S.); ashnywe@uthsc.edu (A.V.N.); wge@uthsc.edu (W.G.)

**Keywords:** Rsr1, Ras, *Aspergillus fumigatus*, polarity establishment, germination

## Abstract

Cell polarization comprises highly controlled processes and occurs in most eukaryotic organisms. In yeast, the processes of budding, mating and filamentation require coordinated mechanisms leading to polarized growth. Filamentous fungi, such as *Aspergillus fumigatus*, are an extreme example of cell polarization, essential for both vegetative and pathogenic growth. A major regulator of polarized growth in yeast is the small GTPase Rsr1, which is essential for bud-site selection. Here, we show that deletion of the putative *A. fumigatus* ortholog, *rsrA*, causes only a modest reduction of growth rate and delay in germ tube emergence. In contrast, overexpression of *rsrA* results in a morphogenesis defect, characterized by a significant delay in polarity establishment followed by the establishment of multiple growth axes. This aberrant phenotype is reversed when *rsrA* expression levels are decreased, suggesting that correct regulation of RsrA activity is crucial for accurate patterning of polarity establishment. Despite this finding, deletion or overexpression of *rsrA* resulted in no changes of *A. fumigatus* virulence attributes in a mouse model of invasive aspergillosis. Additional mutational analyses revealed that RsrA cooperates genetically with the small GTPase, RasA, to support *A. fumigatus* viability.

## 1. Introduction

*Aspergillus fumigatus* is the most common filamentous fungal pathogen in immunocompromised patients and is the causative agent of invasive infections associated with unacceptably high mortality rates [1]. In order to produce invasive infections, resting conidia of *A. fumigatus* must undergo a process of germination whereby the inhaled conidia break dormancy and initiate a highly polarized growth process. This polarized growth process focuses new cell wall and membrane material specifically to the germ tube apex, underpinning the development of hyphae that invade substrate material for nutrient acquisition. Conidial germination and hyphal growth are regulated in response both to extracellular and intracellular stimuli which activate signal transduction pathways that orchestrate the multi-complex polarity machinery by regulating the organization of the cytoskeleton, secretion and endocytosis, recruitment of cytoplasmic factors, and delivery of new membrane components to the polarity site [2]. The study of these molecular mechanisms in pathogenic filamentous organisms such as *A. fumigatus* is expected to deepen our understanding of invasive fungal infections and to potentially lead to the discovery of novel antifungal targets focused on blocking the initiation or halting the progression of tissue invasion.

Signaling networks orchestrated by GTPase proteins are often described as major contributors to germination and polarized growth processes in fungi. As such, signaling pathways headed by members of the Ras superfamily are known to be important for many fungal cellular processes including polarity establishment, asexual and sexual development, stress responses and virulence [3]. RasA is the major Ras sub-family GTPase in *A. fumigatus* and roles for RasA-driven networks in the pathogenic potential of this fungus have been previously studied by our group [4,5,6,7,8,9,10,11,12]. As with the majority of GTPase proteins, RasA activation is positively regulated by Guanine nucleotide Exchange Factors (GEFs), which promote the release of guanosine diphosphate (GDP)-bound GTPases, and negatively regulated by GTPase Activating Proteins (GAPs), which stimulate the intrinsic GTPase activity of these proteins [13]. We recently uncovered a genetic interaction between two RasGEF proteins, GefA and GefB, showing that deletion of both RasGEFs produced a synthetic lethal phenotype in *A. fumigatus* [12]. Although GefA appears to be a direct regulator of RasA in *A. fumigatus*, GefB shares a higher level of homology with Bud5p, a RasGEF that regulates the activity of Rsr1p in the model yeast *Saccharomyces cerevisiae* [12,14,15].

Rsr1 functions have been extensively studied in budding yeast, but little information is available about the function of this GTPase in filamentous fungi. Several studies in *S. cerevisiae* have shown that the Rsr1 GTPase module, formed by Rsr1p together with its activator (Bud5p) and its inactivator (Bud2p), is crucial for correct bud-site selection [15,16,17,18,19,20,21,22]. During yeast cell division, cortical landmark proteins promote localized activation of Rsr1p by Bud5p, which initiates recruitment of the polarization machinery components to the incipient bud site. Rsr1p activation drives the concentration of GTP-Cdc42p at the bud site membrane and subsequently results in symmetry breaking, characterized by a switch from isotropic growth to growth along a polarized axis [19,23,24,25,26,27,28]. Several studies in *S. cerevisiae* have demonstrated that, although Rsr1p itself has no effect on growth rate, deletion or overexpression of Rsr1 causes randomization of the selection of budding sites. Therefore, Rsr1p is essential for normal bud site patterns but not for polarity maintenance or bud formation [16,17]. Similar results are observed in the human pathogen *Candida albicans*. However, in contrast to *S. cerevisiae*, the Rsr1 GTPase module contributes to hyphal tip morphogenesis in *C. albicans* by regulating the distribution of Cdc42 at hyphal tips [29]. A CaRsr1 deletion strain has a random bud site selection pattern but also shows abnormal yeast and hyphal cell morphology, abnormal number of hyphal branches, and loss of viability at 42 °C [30].

In filamentous fungi, where cytokinesis does not result in the complete separation of mother and daughter cells, the role of Rsr1 is still largely unclear. In *Ashbya gossypii*, a fungus closely related to budding yeast, AgRsr1 is important for normal hyphal growth rate, morphogenesis and polarity maintenance. Disruption of this gene causes mislocalization of an important component of the polarisome, Spa2, and this translates into significantly reduced growth rates due to frequent phases of pausing at the hyphal tips, atypical morphology and development of aberrant branching sites [31]. In the filamentous fungus model organism *Neurospora crassa,* disruption of the ScRsr1 homolog, Krev-1, does not affect conidial germination, hyphal growth rates or morphology. However, mutants with constitutively active Krev-1 display an inhibition in the development of perithecia, suggesting that this gene is important for sexual cycle progression [32].

Here, we explore roles for the Rsr1 ortholog, RsrA, in hyphal growth and germination of *A. fumigatus*. Whereas deletion of *rsrA* resulted in reduced growth and germination rates accompanied by normal hyphal morphology, overexpression of *rsrA* caused increased instances of polarity establishment sites during germination. Despite this finding, neither loss- nor gain-of-function RsrA mutations altered the virulence potential of *A. fumigatus*. We also report that, similar to our earlier studies with the putative RasA and RsrA GEF proteins, GefA and GefB, double deletion of the putative downstream GTPases, RasA and RsrA, is synthetically lethal.

## 2. Materials and Methods

### 2.1. Strains and Growth Conditions

The *A. fumigatus* strains used in this study are listed in Table 1. All strains were cultured in Glucose Minimum Medium (GMM) agar [33] plus uridine/uracil, when necessary, and incubated at 30 °C or 37 °C for 4–7 days. To assess fungal morphology, strains were cultured in filtered GMM broth for 14–18 h at 30 °C or 37 °C. To culture strains for genomic DNA or total RNA extraction, GMM was supplemented with 0.5% yeast extract (GMM-YE) in order to potentiate growth. GMM supplemented with 1.2 M sorbitol (Sorbitol Minimum Medium (SMM)) was utilized after fungal transformations, to allow the protoplasts to recover properly.

### 2.2. Analysis of Growth and Germination Rates

To determine if RsrA is important for *A. fumigatus* growth, five microliters containing 10^4^ conidia were inoculated in the center of GMM agar plates and incubated at 37 °C, and diameters were measured every day for a 96 h period. Pictures were taken at the end of day 4. For germination assays, four thousand conidia were inoculated in GMM broth and at least five hundred conidia were counted every hour. The presence of a germ tube was considered to be the endpoint to define establishment of polarity. The involvement of RsrA in germ tube emergence orientation was also evaluated. To do this, conidia were incubated in GMM at 37 °C until two germ tubes were observed and the germlings were scored for right/acute (≤90°) or obtuse (>90°) angle of germ tube orientation. Three biological replicates were performed for each analysis, and data are presented as mean ± standard deviation.

### 2.3. Stress Analyses

To determine potential roles for RsrA in response to osmotic, membrane and cell wall stresses, spot dilution assays were performed. Briefly, suspensions of 10 to 10^4^ conidia were inoculated in GMM supplemented with 1 M NaCl, 1 M KCl, 1.2 M sorbitol, 0.0125% SDS, Congo Red (40 and 80 µg/mL) or Calcofluor White (40 and 80 µg/mL). Agar plates were incubated at 37 °C for 72 h. To evaluate susceptibility to oxidative stress, GMM agar plates were inoculated with 1 × 10^7^ conidia and allowed to dry. Then, a section of the center of the agar was excised and filled with 50 µL of a solution of 200 mM H_2_O_2_. Inhibition zone diameters were measured after 24 h at 37 °C. Caspofungin Minimal Effective Concentration (MEC) and voriconazole Minimum Inhibitory Concentration (MIC) were determined at 24 and 48 h, respectively, as recommended in the CLSI M38 document (CLSI 2017). All analyses were performed in triplicates, and data are represented as mean ± standard deviation.

### 2.4. Genetic Manipulations

The *rsrA* deletion mutant was generated by replacing the complete *rsrA* Open Reading Frame (ORF) by the *Aspergillus parasiticus pyrG* cassette [35] in the *A. fumigatus* KU80Δ*pyrG* genetic background [34]. This strain is unable to grow in the absence of uridine/uracil, because it is deficient in orotidine 5′-monophosphate decarboxylase activity. Therefore, the resulting *rsrA* deletion is pyrimidine prototrophic. Protoplast generation and transformation were performed as previously described [36] and colonies able to grow in media lacking uridine/uracil were genotyped by multiple PCRs of genomic DNA.

For complementation of the deletion mutant, *rsrA* was ectopically expressed under the control of its native promoter and tagged with Green Fluorescent Protein (GFP) at the N-terminus for subsequent localization studies, using our CRISPR-Cas9 technique [37]. Briefly, the *rsrA* coding sequence and approximately 1 kb of the *rsrA* promoter were cloned into plasmid pAGRP, which contains enhanced GFP (eGFP) and a phleomycin resistance cassette [5]. The repair template for CRISPR/Cas9-based integration was amplified from the modified pAGRP and contained 35 bp regions of microhomology targeting either side of protospacer adjacent motifs (PAMs) both up- and downstream of the non-functional *A. fumigatus pyrG* locus, as previously described [12].

To decipher subcellular localization of RsrA in the absence of RasA, a *rasA* deletion was generated in the Δ*rsrA* + GFP-*rsrA* strain as well as in the Δ*akuB-pyrG^+^* control strain. Briefly, two PAMs were first chosen, one upstream and one downstream of the *rasA* coding sequence, to direct Cas9-induced double strand breaks and mediate repair template integration, as we have previously described [37]. A hygromycin resistance repair template was amplified from plasmid pJMR2 [38] and utilized in a CRISPR/Cas9-mediated transformation to replace the entire *rasA* locus, generating a complete gene deletion [37].

Two mutant strains with regulatable *rasA* expression were generated by replacing the *rasA* native promoter with the tetracycline-inducible pTetOn promoter in both the Δ*akuB-pyrG^+^* and Δ*rsrA* genetic backgrounds. A repair template containing a phleomycin resistance gene followed by the TetOn cassette was amplified from the plasmid pCH008-phleo to contain 35 bp regions of microhomology to the *rasA* promoter region immediately upstream of the *rasA* transcriptional start, as recently performed [12]. Protoplasts were plated on SMM supplemented with 62 µg/mL of phleomycin and 30 µg/mL of doxycycline. Further PCR analyses were used to confirm the correct promoter replacement.

For *rsrA* overexpression, the endogenous *rsrA* promoter was replaced by the *A. fumigatus* heat shock protein A (*hspA*) promoter in the Δ*akuB-pyrG^+^* genetic background [39]. Briefly, a repair template containing a hygromycin resistance cassette followed by the *hspA* promoter was amplified from plasmid pJMR2 [38] and contained microhomology regions homologous to the promoter region of *rsrA*. In this case, a PAM site was chosen in the position −4 bp relative to the *rsrA* start codon.

All transformations were performed as previously described [37] and hygromycin- or phleomycin-resistant colonies were screened and genotyped by multiple PCRs to confirm a precise homologous integration.

### 2.5. Fluorescence Microscopy

To analyze the GFP-*rsrA* strains for RsrA subcellular localization, 10^3^ conidia were inoculated onto sterile coverslips in 5 mL of GMM and incubated in static culture at 37 °C for 14 h. The coverslips were then washed with distilled water and GFP fluorescence was analyzed using a Nikon Ni-U fluorescence microscope equipped with a Nikon DS-Qi1 Mc camera using a GFP filter. Images were captured using Nikon Elements software (v. 4.0). Images were then processed using the Nikon Elements AR Analysis 2-D Deconvolution software on Automatic settings. To assess the effect of cytochalasin A, 10^3^ conidia from both control and Δ*rsrA* + GFP-*rsrA* strains were grown as described above. After 14 h, the medium was removed and replaced with fresh media containing 10 µg/mL of the anti-actin agent and incubated for two additional hours. The strains were finally observed under a fluorescence microscope, as described above.

### 2.6. RNA Extraction and RT-qPCR

For RNA extraction, 5 × 10^7^ conidia were inoculated in GMM-YE and grown for 18 h at 30 °C or 37 °C with agitation at 250 rpm. After this time, the mycelia were washed with distilled water and crushed with liquid nitrogen. The powdered samples were resuspended in 1 mL of Trizol (Invitrogen^TM^, Waltham, MA, USA) and incubated on ice for 5 min. Afterwards, 0.2 mL of chloroform-isoamyl alcohol (24:1) were added, followed by 3 min on ice. After centrifugation of 15 min at 13,000 rpm, the clear aqueous phase was mixed with an equal volume of ice-cold 70% ethanol and the RNA extraction was continued using an RNAeasy kit (Qiagen, Germantown, MD, USA). Six micrograms of RNA were treated with Turbo DNAase (Invitrogen, Carlsbad, CA, USA) following the manufacturer’s instructions. Five hundred nanograms of RNA were retrotranscribed using the ProtoScript II cDNA synthesis kit (New England Biolabs, Ipswich, MA, USA) and the resulting cDNA was treated with RNAase H (New England Biolabs, Ipswich, MA, USA) for 20 min at 37 °C to remove any remaining RNA traces. RT-qPCR was performed in 20 µL PCR reactions in a CFX Connect Real-Time System (Bio-Rad, Hercules, CA, USA) using SYBR^®^ Green Master Mix (Bio-Rad, Hercules, CA, USA). The conditions were as follows: 95 °C for 3 min, followed by 40 cycles of 10 s at 95 °C, 15 s at 55 °C and 45 s at 72 °C. A melting curve analysis was carried out immediately after PCR completion to check specificity of the amplification. Primers specific for RsrA and β-tubulin A (TubA) were designed to flank introns. Cycle threshold (Ct) values for RsrA were normalized to those of the housekeeping TubA and the relative expression was determined using the 2^−ΔCt^ and 2^−ΔΔCt^ methods [40]. Each analysis was performed in technical and biological triplicates.

### 2.7. Murine Model of Invasive Pulmonary Aspergillosis

Six-week-old CD-1 female mice (Charles River, Wilmington, MA, USA), weighing approximately 25 g, were immunosuppressed intraperitoneally with 150 mg/kg of cyclophosphamide starting 3 days before the infection and on days 1, 4 and 7 post-inoculation. In addition, a single injection of triamcinolone acetonide (Kenalog, Bristol-Myers Squibb, Princeton, NJ, USA) was administered subcutaneously the day before the inoculation. On day 0, the mice were slightly anesthetized with isoflurane and challenged with 10^6^ conidia suspended in saline solution, by nasal instillation. The animals were monitored at least twice a day for a period of two weeks and those showing severe signs of disease were humanely euthanized by anoxia with CO_2_. In order to prevent bacterial infections, mice were given a mixture of sulfamethoxazole and trimethoprim in their drinking water, starting 3 days before the infection. Survival curves were compared using the log-rank test in GraphPad Prism v. 8.2.1 for Windows. The studies were performed in accordance with the approved protocol (project identification code: 19-0067.0, date of approval: 07/2020) by the Institutional Animal Care and Use committee of the University of Tennessee Health Science Center.

## 3. Results

### 3.1. RsrA Regulates Growth and Polarity Establishment in A. fumigatus

To identify an *A. fumigatus* homolog of the *S. cerevisiae* RsrA protein, a BLAST search was performed against the *A. fumigatus* Af293 genome (FungiDB; Genome Version 2015-09-27) using the *S. cerevisiae* protein sequence from the Saccharomyces genome database. The protein putatively encoded by Afu5g08950 shares a predicted amino acid sequence identity of 55.56% with the *S. cerevisiae* RsrAp. This putative protein is 210 amino acids in length and encodes an Ras effector domain as well as GTP/GDP binding domains and a CAAX box (Appendix A). We have therefore named this putative *A. fumigatus* protein RsrA. A notable difference between the yeast and *A. fumigatus* RsrA sequences is found in the length of the hypervariable region, with the *A. fumigatus* RsrA region being shorter (Appendix A). This finding is similar to previously reported differences in the hypervariable regions of *A. fumigatus* RasA and *S. cerevisiae* Ras2p [7].

To analyze the cellular functions of this small GTPase in *A. fumigatus*, we first generated a gene deletion strain and, to ensure that the resulting phenotypes were due to the specific deletion of *rsrA*, a complemented strain was also generated by ectopic integration of a GFP-fused *rsrA* allele under the control of the native *rsrA* promoter. This strain was referred to as Δ*rsrA* + GFP-*rsrA*. The strain Δ*akuB*-*pyrG*^+^ was used as a control for all the assays performed in this study [11].

When the macro-and micro-morphology of the control and Δ*rsrA* strains were analyzed, we observed no significant differences, suggesting that RsrA does not have an overt role in *A. fumigatus* hyphal morphogenesis (Appendix A). However, growth and germination (i.e., establishment of a polarized growth axis) rates in glucose minimal media (GMM) were impaired when the GTPase was deleted (Figure 1). We observed a statistically significant reduction of growth after 48, 72 and 96 h of incubation in GMM agar at 37 °C (Figure 1A,B). This reduction in growth was accompanied by a delay in the initiation of germination of approximately two hours (Figure 1C). Five percent (±1%) of conidia from the control strain developed a germ tube after 5 h of incubation in GMM at 37 °C, whereas conidia of the deletion mutant initiated polarity establishment after 7 h. Further, almost 90% (88 ± 1.73%) of conidia from the control strain had germinated after 8 h, whereas only 39% (±4.6%) of conidia from Δ*rsrA* had formed a germ tube by this time point. To determine whether RsrA activity impacted the pattern of germ tube emergence, the isogenic strain set was cultured in GMM at 37 °C until 100% of the germinating conidia had generated two germ tubes. Five hundred germlings from each strain were then scored for obtuse (>90°) or right/acute (≤90°) angles of orientation of the two initial germ tube events, as shown in the schematic representation in Figure 1D. After 12 h of growth at 37 °C, almost 100% of the wild type germlings had generated two germ tubes, whereas the *rsrA* deletion mutant required two additional hours to obtain the same result. We observed that 95.4% (±1.8) of control and 94% (±2.3) of Δ*rsrA* germlings formed germ tubes at >90° angle, and 4.6% and 6% formed germ tubes at ≤90° angle, respectively (Figure 1D). Therefore, no significant differences were observed between strains. The complemented strain displayed equivalent colony diamaters and patterns of germination as the control strain (Figure 1), indicating that the phenotypes observed were due to the deletion of *rsrA*.

To identify potential roles for RsrA in *A. fumigatus* stress responses, Δ*rsrA* was also subjected to diverse external stressors, including: cell wall stress, using calcofluor white and congo red, compounds that target chitin and glucans, respectively; osmotic stress, using NaCl, KCl and sorbitol; and oxidative stress employing hydrogen peroxide, in a spot dilution assay. When subjected to cell wall and oxidative stresses, the Δ*rsrA* mutant displayed growth similar to the control and complemented strains (Appendix A). However, a slight increase in susceptibility of the knockout strain was observed in the presence of osmotic stress driven by NaCl or KCl, but not in sorbitol, and surprisingly, this phenotype was only partially restored in the complemented strain (Appendix A). This partial complementation could be the result of interefence by the N-terminal GFP-tag or the transcriptional overexpression of *rsrA*, resulting in tagging *rsrA* with GFP in comparison to the control strain (Appendix A). Additionally, in vitro susceptibility testing to echinocandin and triazole class antifungals was performed. We observed a caspofungin MEC of 0.125 µg/mL and a voriconazole MIC of 0.25 µg/mL for control, Δ*rsrA* and complemented strains. Additional phenotypic screens included the evaluation of growth at different temperatures (30, 37 and 42 °C). However, no significant differences in growth or germination potential were observed (data not shown).

### 3.2. Overexpression of RsrA Alters the Spatial Regulation of Polarity Establishment

When *rsr1* is overexpressed in budding yeast, dividing cells show the same random budding pattern as when the gene is deleted [16,17]. Therefore, we overexpressed *rsrA* by replacing the endogenous promoter with 1000 bp of the *A. fumigatus* Heat Shock Protein A (*hspA*) promoter (P_hspA_), using a CRISPR-Cas9 gene targeting technique (Figure 2A). This promoter has been previously shown to drive high level expression of downstream genes in a temperature-dependent manner when used for ectopic gene expression [38,39]. Transcriptional profiling of the P_hspA_-*rsrA* and control strains, analyzed by RT-qPCR at 30 °C or 37 °C, revealed that expression of *rsrA* was increased 22-fold or 46-fold, respectively (Figure 2B). Germination assays at 37 °C revealed that conidia of the *A. fumigatus* control strain started to establish polarity axes after 5 h of incubation in liquid GMM, and 100% of conidia had generated at least one germ tube within 12 h of culture (Figure 2C). However, high-level overexpression of *rsrA* (37 °C culture) significantly delayed germination. Initial germ tube development in the P_hspA_-*rsrA* mutant was only observed after 12 h of culture at 37 °C and approximately 90% of conidia had germinated after 26 h (Figure 2C). At the end of 26 h of incubation at 37 °C, we also observed that 9.7% (±0.6) of the P_hspA_-*rsrA* conidia were only able to undergo the initial isotropic swelling phase of germination and did not establish a polarized growth axis. When *rsrA* expression levels were decreased by reducing the culture temperature to 30 °C, 5.7% (±1.5%) of the conidia from the control strain had generated the first germ tubes after 10 h, whereas germination in the *rsrA* overexpression strain was delayed by only one hour (Figure 2C). After 18 h of growth in GMM at 30 °C, 100% of conidia from the control strain and 97.7 ± 1.52% of the P_hspA_-*rsrA* strain had successfully established polarity and generated at least one germ tube. Although high-level overexpression of *rsrA* generated a significant polarity establishment defect, this delay in polarity establishment was not accompanied by a decreased colony diameter. On the contrary, an increase in the colony diameter after 72 and 96 h of growth in GMM at 37 °C was observed for the P_hspA_-*rsrA* mutant (Figure 2D). These data suggested that, although initial polarity establishment is delayed, overexpression of *rsrA* may promote rapid hyphal growth rate in *A. fumigatus*.

Notably, the P_hspA_-*rsrA* strain also displayed an atypical micro-morphology. During early germination events at 37 °C, conidia from the control strain grew isotropically for 5–6 h, switching after this incubation period to polarized growth with only two polarized growth axes generated (Figure 3A). In contrast, conidia of the overexpression mutant exhibtied exaggerated swelling during the initial 12 h of culture, reaching a size of about 25 µm, approximately five times larger than swollen conidia of the control strain (Figure 3A). Once germinating conidia of the P_hspA_-*rsrA* mutant became competent for polarity establishment, a significantly increased number of polarity axes was noted within individual conidia (Figure 3A). The morphology of polarity establishment was strikingly different when the P_hspA_-*rsrA* strain was cultured at 30 °C, displaying development of normal germination patterns and conidial morphology (Figure 3B). As shown in Figure 3B, after 14 h of incubation, the size of the swollen conidia from the *rsrA* overexpression mutant appeared more similar to the control at 30 °C. In addition, only one germ tube was generated at a time and only two to three total germ tubes were produced by each germinating conidium. The control strain displayed the same morphology at both temperatures tested (Figure 3A,B). These results suggest that RsrA acts as a spatial regulator of polarity establishment and that accurate expression levels are essential to support normal germination patterns.

### 3.3. A. fumigatus RsrA Localizes to Septa

Previous studies have shown that, in *S. cerevisiae*, Rsr1p is widely distributed intracellularly but it is enriched at the plasma membrane and at sites of polarization [24]. Similarly, a GFP-AgRsr1p fusion protein localizes to the hyphal tip in *A. gossypii* [31]. As deletion and overexpression of *rsrA* suggested a role in polarity establishment and hyphal growth rate, we next sought to determine the sub-cellular localization of RsrA in *A. fumigatus* using the Δ*rsrA* + GFP-*rsrA* complemented strain. Importantly, as evidenced by the phenotypic data outlined above, the strain expressing GFP-*rsrA* was morphologically indistinguishable from the control strain, suggesting that the fusion protein is functional. During mature hyphal growth, the fluorescence pattern of GFP signal revealed that the GFP-*rsrA* chimeric protein was localized throughout the cytoplasm and was also concentrated at septa (Figure 4A). To see if localization of RsrA to septa was dependent on active actin polymerization, we employed treatment with cytochalasin A. This compound is an actin polymerization inhibitor that binds to actin filaments ends and, consequently, limits actin assembly [41]. The control and the Δ*rsrA* + GFP-*rsrA* strains were cultured for mature hyphal development in media lacking cytochalasin A, and then treated with the anti-actin agent for two additional hours of culture. As shown in Figure 4B, treatment with cytochalasin A was associated with a reduction in septum fluorescence and induced cytoplasmic mislocalization of RsrA (Figure 4B). Control experiments with DMSO did not show any detectable alteration when compared to the control without the solvent (Figure 4C). These results suggest that RsrA presence at the septum is dependent on an intact actin cytoskeleton.

### 3.4. RsrA Is Dispensable for A. fumigatus Virulence

Because overexpression of *rsrA* caused altered germination at host physiological temperatures, we next sought to determine whether our loss- or gain-of-function mutations would influence murine invasive pulmonary aspergillosis outcomes. In order to examine the impact of these polarity establishment defects on virulence, we used a chemotherapeutic murine model of invasive pulmonary aspergillosis, and virulence was assessed using survival as the endpoint. Regardless of experimental arm, animals started to succumb to the infection 4 days after the inoculation. We observed that neither deletion nor overexpression of *rsrA* altered virulence of *A. fumigatus* (Figure 5). In fact, these strains caused an 87.5% (7 out of 8 mice) of mortality by day 8 post-infection, whereas 37.5% (3 out of 8 mice) of animals inoculated with the control strain were alive by the end of the experiment. However, no statistical differences were observed between survival curves. The median survival times for Δ*akuB-pyrG*^+^, Δ*rsrA*, P_hspA_-*rsrA* and Δ*rsrA* + GFP-*rsrA* groups were 5.5, 6, 5.5 and 6.5 days, respectively. These results suggest that RsrA is not required for the ability of *A. fumigatus* to invade tissue and generate disease in the host.

### 3.5. RsrA and RasA Interact Genetically to Control Polarity Establishment

Previous studies in yeast have shown that the major Ras sub-family proteins, Ras1p/Ras2p, and Rsr1p have overlapping roles supporting viability, as the loss of Ras1/2p and Rsr1p activities in *S. cerevisiae* and *Y. lypolitica* is lethal [42,43]. To determine whether a similar relationship between RsrA and RasA pathways exists in filamentous fungi, we attempted to generate a double deletion mutant. Viable transformants were unobtainable after multiple attempts, suggesting that these pathways may share an overlapping role essential to *A. fumigatus* viability. To confirm this, we next constructed a strain with regulatable *rasA* expression by replacing the *rasA* promoter with the tetracycline-inducible TetOn promoter [44] in the Δ*rsrA* background. This new strain was referred to as Δ*rsrA*/pTetOn-*rasA*. To generate a reliable control strain confirming phenotypic results of repression of *rasA* expression, we performed the same genetic modification in the control strain Δ*akuB*-*pyrG*^+^ (Figure 6A). This control strain was referred to as pTetOn-*rasA*. When culturing the pTetOn-*rasA* control strain on GMM in the absence of doxycycline, we observed the production of a compact colony morphology that was associated with stunted, swollen hyphae characteristic of the Δ*rasA* strain reported previously (Figure 6B,C, Appendix A and [6]). In contrast, in the presence of increasing doxycycline concentrations, the macroscopic colony phenotype of the pTetOn-*rasA* mutant improved and was indistinguishable from that of the parental strain at 30 µg/mL doxycycline (Figure 6B). Hyphal morphology of pTetOn-*rasA* in submerged GMM broth culture also improved in the presence of increasing doxycycline concentrations, with hyphae maintaining stable growth axes and resembling wild type growth at 60 µg/mL doxycycline (Figure 6C, top row panels, and Appendix A). Because the pTetOn-*rasA* mutant phenocopied the Δ*rasA* strain under non-inducing conditions (Appendix A), we concluded that effective suppression of *rasA* expression was achieved in the absence of doxycycline. In contrast, when *rasA* expression was suppressed in the Δ*rsrA* background (Δ*rsrA*/pTetOn-*rasA*, 0 µg/mL doxycycline), the majority of the conidia underwent isotropic growth but were not able to establish polarity after 24 h of growth in GMM at 37 °C (Figure 6C, bottom panels). Increasing concentrations of doxycycline overcame this growth defect by inducing normal polarity establishment rates and hyphal morphology (Figure 6C, bottom panels). Wild type morphology was generated by addition of 120 µg/mL doxycycline to the GMM broth culture. These results support the hypothesis that RsrA and RasA interact genetically during early growth events.

### 3.6. RsrA Gene Expression, but Not Protein Localization, Is Dependent on RasA

We have shown that RsrA localization at septa is actin-dependent and that RsrA and RasA genetically interact to orchestrate polarity establishment. Fungal Ras orthologs are also known to regulate the polarization of the actin cytoskeleton [45,46,47,48,49,50,51], therefore we next investigated whether RsrA localization in *A. fumigatus* might be dependent on the presence of RasA. To do this, we generated a complete deletion of the *rasA* ORF in the background of the Δ*rsrA* + GFP-*rsrA* strain. The resulting mutant strain displayed the same stunted colony and hyphal morphology as the *rasA* deletion mutant. Although the morphology and phenotype of the RasA knockout has been well characterized in our laboratory [6], for better comparison purposes in the present study, we generated a *rasA* deletion strain in the Δ*akuB*-*pyrG*^+^ genetic background that was completely identical to the Δ*rasA* previously characterized by our group (Appendix A and [6]).

As shown in Figure 7A and as described above, RsrA is localized throughout the cytoplasm and concentrated at septa in the Δ*rsrA* + GFP-*rsrA* strain. When *rasA* was deleted in this genetic background (Δ*rsrA* + GFP-*rsrA*/Δ*rasA*), the GFP-*rsrA* signal remained at septa but also displayed an intense, diffuse cytoplasmic fluorescence when compared to a wild type background (Figure 7A). This surprising result suggested that GFP-*rsrA* presence might be increased when RasA activity is lost, likely either through increased *rsrA* gene expression or stabilization of RsrA protein. To test whether *rsrA* gene expression levels may be induced by the loss of *rasA*, the expression of *rsrA* was determined using RT-qPCR. Indeed, the Δ*rsrA* + GFP-*rsrA*/Δ*rasA* strain expressed 2.5-fold more *rsrA* than the parent Δ*rsrA* + GFP-*rsrA* strain (Figure 7B). Taken together, these findings suggest that RasA regulates RsrA at the level of gene transcription, and presumably, at the protein level for correct subcellular localization.

## 4. Discussion

In this study, we report the characterization of the *A. fumigatus* RsrA small GTPase, orthologs of which are known mediators of polarized growth in yeast. However, our results surprisingly showed that RsrA is preferentially localized to the septum in *A. fumigatus* and not to the hyphal tip, as described for *A. gossypii* [31]. We found that the presence of this protein is non-essential for *A. fumigatus* growth, for the proper selection of growth axes during initial polarity establishment occurring after spore germination, or for normal branching patterns (Appendix A). In contrast, as overexpression of *rsrA* resulted in aberrant germination phenotypes, we conclude that proper regulation of RsrA activity levels is likely necessary for spatiotemporal control of early events in *A. fumigatus* growth. The current study is only the third reported functional analysis of an Rsr1 ortholog in a monomorphic filamentous fungus, the previous two being for the Saccharomycete mold, *A. gossypii*, and for the Pezizomycotina mold, *Neurospora crassa* [31,32]. Whereas *A. gossypii* AgRsr1p guides directional hyphal growth via stabilization of the polarisome at the hyphal tip, mutation of the *N. crassa* Rsr1 ortholog, krev-1, produces no aberrant vegetative growth phenotypes [31,32]. Instead, the primary function of the Krev-1 GTPase in *N. crassa* appears to be the regulation of sexual cycle progression [32]. Differences between basal phenotypes of Rsr1 ortholog mutants in these filamentous fungi show that, although the general GTPase modules may be conserved, their functional output varies.

Strikingly, the polarity establishment defects of the *rsrA* overexpression mutant were only evident at high levels of over-expression, induced by host-physiological temperatures. Despite this finding, virulence in a murine model of invasive aspergillosis was unaltered by mutation of *rsrA*. A previous study with *C. albicans* revealed that CaRsr1 is important for virulence in a gene-dose dependent manner. *C. albicans* is the only other human pathogenic fungus in which an Rsr1 ortholog has been studied. Whereas a heterozygote CaRSR1/Carsr1 strain was reported to cause 60% mortality, a homozygote disruptant strain was completely avirulent in a neutropenic murine model of disseminated candidiasis [52]. *C. albicans* is a pleomorphic yeast that grows in budding yeast, pseudohyphal and hyphal forms, and the transition of yeast-to-hyphae growth is a known virulence determinant. Disruption of CaRSR1 negatively impacts each of these growth forms, generating a mutant with reduced pathogenic fitness. Although *A. fumigatus rsrA* overexpression did result in a temperature-sensitive germination defect, the pathogenic potential of the P_hspA_-*rsrA* strain was likely maintained by the normal hyphal morphology and rapid growth rate of this mutant. Therefore, *A. fumigatus* RsrA signaling plays no overt role in tissue invasion and virulence.

Although overexpression of *rsrA* resulted in a significant delay in the establishment of polarity, breaking of conidial dormancy during germination was not delayed. This fact was evidenced by the development of large, isotropically swollen conidia in this mutant. Interestingly, we reported a similar phenotype in a mutant strain expressing constitutively active RasA in *A. fumigatus* [7,9]. When RasA protein activity cannot be properly regulated due to the introduction of a point mutation that inhibits the RasA–RasGAP interaction, *A. fumigatus* conidia undergo a prolonged period of isotropic swelling before polarity is eventually established [7,9]. However, unlike the *rsrA* overexpression strain, the pattern of germ tube emergence is not altered by constitutively active RasA [7,9]. Nevertheless, these results suggest co-regulation of RasA and RsrA pathways wherein activity of both RasA and RsrA must increase for the efficient breaking of spore dormancy and then be properly downregulated for the correct timing of polarity establishment. However, RsrA also plays a separate role in the spatial positioning of new growth axes during initial polarity establishment. Previous studies in yeast revealed a genetic interaction between these signaling pathways, as disruption of Ras1, Ras2 and Rsr1 results in inviability [42]. In the yeast *Yarrowia lipolytica*, deletion of Rsr1 in cells lacking Ras1 also results in lethality [43]. As with yeast, we found loss of both *rsrA* and *rasA* to be synthetically lethal and, in *A. fumigatus*, this lethality is likely due to the inability of the double mutant to establish a polarized growth axis. In addition, we have revealed novel links between these pathways in *A. fumigatus*, as loss of RasA caused both increased expression of *rsrA* while not affecting the localization of RsrA to septa.

The precise mechanism underlying how RsrA- and RasA-driven pathways work in concert to orchestrate the early events in *A. fumigatus* growth are still under investigation. Several studies have demonstrated that the Rsr1 GTPase module interacts genetically and physically with Cdc42 and its regulators to organize the actin cytoskeleton and septin filaments to the selected site of polarized growth in yeast [16,19,23,25,26,28,53,54,55]. Active Rsr1 recruits the GEF, Cdc24, to accomplish Cdc42 activation at the incipient bud site in yeast [23,54]. Cyclical inactivation of Rsr1 results in Cdc42 release, ensuring the integrity of yeast phase growth is maintained. For organisms such as *C. neoformans*, in which an Rsr1 ortholog does not exist, Ras proteins have been posited to play an analogous role by interacting with Cdc24 when activated and, in turn, recruiting and activating Cdc42 [56]. Previous studies in not only *C. neoformans*, but in multiple other fungi, have also shown that Ras cooperates with the highly conserved Rho-GTPase Cdc42 to control actin cytoskeleton polarization during growth and differentiation [17,49,50,51,57,58,59,60]. Cdc42 is highly conserved in the eukaryotic kingdom and represents an essential component of the polarisome in both yeast and filamentous fungi. In *A. fumigatus* and other ascomycete molds, both Rsr1 and Ras orthologs exist. Taken together, our previously published and current data support the hypothesis that, at least for *A. fumigatus*, RasA may signal to the Cdc42 module through Rsr1-dependent mechanisms for polarity establishment and through Rsr1-independent pathways for hyphal morphogenesis. Testing this hypothesis will be the focus of future studies.

In summary, the data presented here suggest that, as with *S. cerevisiae* Rsr1, *A. fumigatus* RsrA likely functions as an internal polarity cue for polarized growth, but that this role is largely restricted to initial polarity establishment events during germination. A continued role for polarity maintenance in terms of hyphal growth guidance is less apparent in *A. fumigatus*, as hyphal morphogenesis of the *rsrA* mutants generated here is not significantly altered.

## Figures and Tables

**Figure 1 jof-06-00285-f001:**
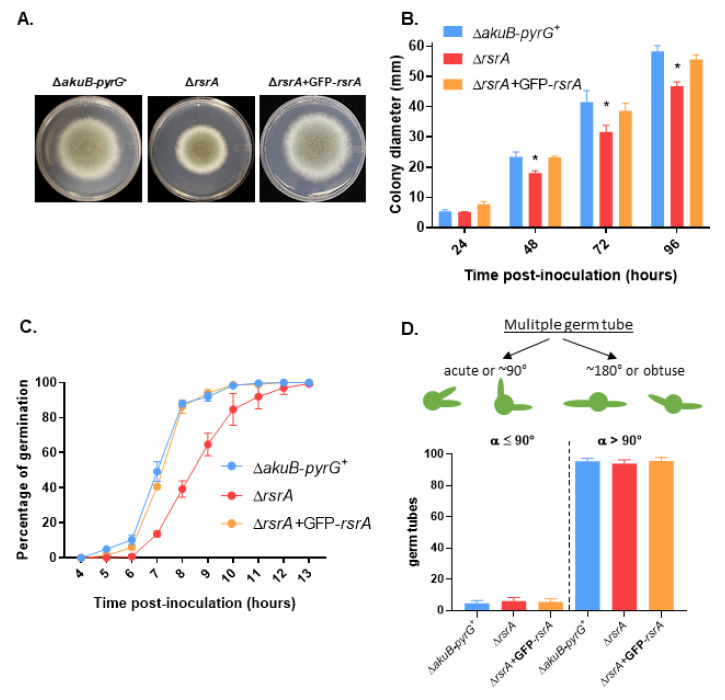
Deletion of *rsrA* causes decreased colony development and delayed polarity establishment. (**A**) Colony morphology of the control (Δ*akuB*-*pyrG*^+^), *rsrA* deletion (Δ*rsrA*) and complemented (Δ*rsrA* + GFP-*rsrA*) strains. Glucose Minimum Medium (GMM) agar plates were point-inoculated in the center of the plate with 5 µL containing 10^4^ conidia and incubated at 37 °C for 96 h.(**B**) Colony diameters were measured every 24 h during a 96 h period. Diameters between strains were compared at each time point using 2-way ANOVA with Tukey’s test for multiple comparisons (GraphPad Prism v. 8.2.1). Asterisks indicate a statistically significant difference (adjusted *p* value < 0.0001) between the Δ*rsrA* and both the control and the complemented strains at the indicated time point. (**C**) Ten thousand conidia were inoculated in GMM broth and incubated at 37 °C. The presence of a germ tube was considered to be the endpoint for polarity establishment, and percentage of germination was scored as the number of conidia producing a visible germ tube amongst 100 enumerated conidia. (**D**) Ten thousand conidia were allowed to grow in GMM at 37 °C until two germ tubes were developed. Then, the orientation of the germ tubes was evaluated for at least 500 conidia and the number of conidia with germ tubes forming an acute/right, or an obtuse angle were quantified and compared between strains using one-way ANOVA. Measurements and error bars represent the mean and standard deviation of 3 independent experiments.

**Figure 2 jof-06-00285-f002:**
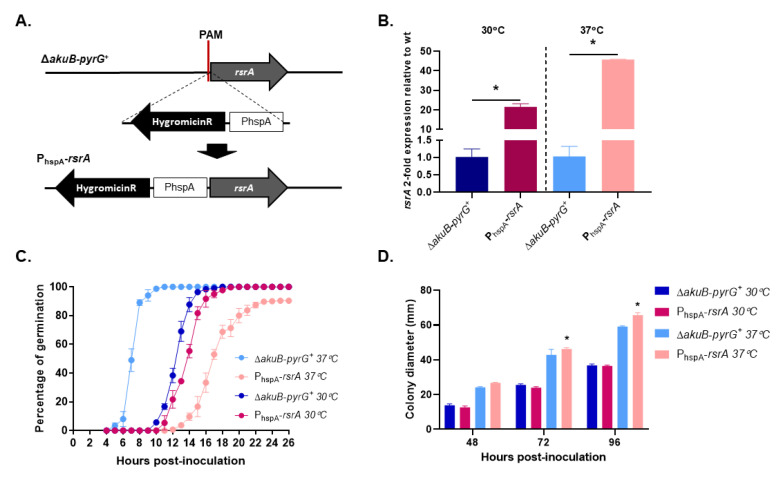
Overexpression of *rsrA* causes a significant delay in polarity establishment. (**A**) Schematic representation of *rsrA* promoter replacement by the constitutive Heat Shock Protein A (*hspA*) promoter. A construct containing the hygromycin resistance cassette together with one thousand base pairs of the *hspA* promoter was used to target a Protospacer Adjacent Motif (PAM) in the *rsrA* promoter. Arrows indicate the direction of gene transcription. (**B**) Expression of *rsrA* was determined by RT-qPCR for both control and overexpressed *rsrA* strains. Cycle threshold (Ct) values of *rsrA* were normalized to those of the endogenous standard *tubA*. Statistical analysis was performed using unpaired, two-tailed, Student’s *t*-test in GraphPad Prism v. 8.2.1 (*p* ≤ 0.0024). (**C**) Polarity establishment is delayed 8 h when *rsrA* is overexpressed. Germination rates of control strain expressing native and overexpressed *rsrA* were determined as explained in Materials and Methods. (**D**) Overexpression of *rsrA* causes an increase in growth rate at 37 °C. Ten thousand conidia were point inoculated in the center of a GMM agar plate and incubated at 30 °C or 37 °C. Colony diameters were measured every 24 h for a 96 h period and compared among groups at each time point using 2-way ANOVA with Tukey’s test for multiple comparisons (GraphPad Prism v. 8.2.1). Asterisks indicate a statistically significant difference (adjusted *p* value < 0.0001) between the control and the P_hspA_-*rsrA* strains at the indicated time point.

**Figure 3 jof-06-00285-f003:**
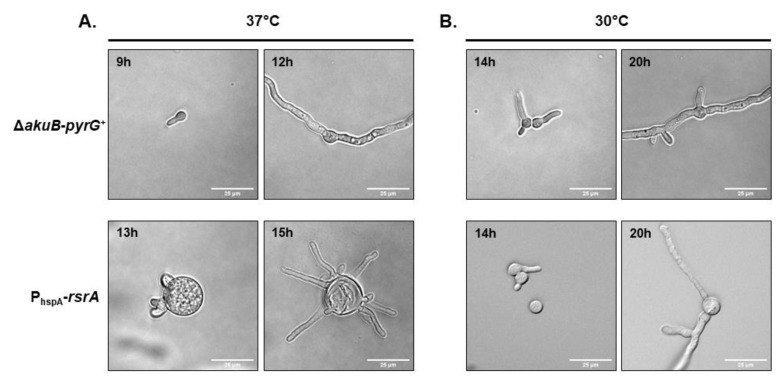
Overexpression of *rsrA* produces a defect in patterning of polarity establishment. Depicted are time course microphotographs showing the development of control and P_hspA_-*rsrA* strains. Two thousand conidia were cultured in liquid GMM and incubated at 37 °C (**A**) or 30 °C (**B**) and microphotographs were taken at the indicated time points.

**Figure 4 jof-06-00285-f004:**
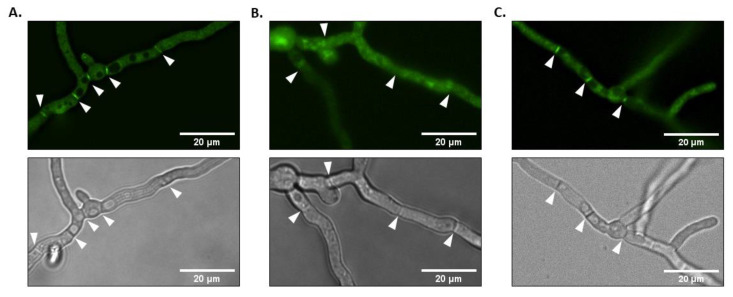
GFP-*rsrA* is concentrated at septa in an actin-dependent fashion. (**A**) One thousand conidia of the Δ*rsrA* + GFP-*rsrA* strain were incubated in GMM broth for 14 h at 37 °C. B. GFP-*rsrA* localization is actin-dependent. One thousand conidia were cultured in GMM for 14 h at 37 °C. The medium was then replaced by fresh GMM containing 10 µg/mL of cytochalasin A (**B**) or DMSO (**C**) and cultures were incubated for an additional 2 h. Note that the addition of 2% DMSO had no effect on hyphal development or localization of GFP-*rsrA*. Microphotographs were taken with a Nikon X fluorescence microscope equipped with a GFP filter. White arrows indicate septa.

**Figure 5 jof-06-00285-f005:**
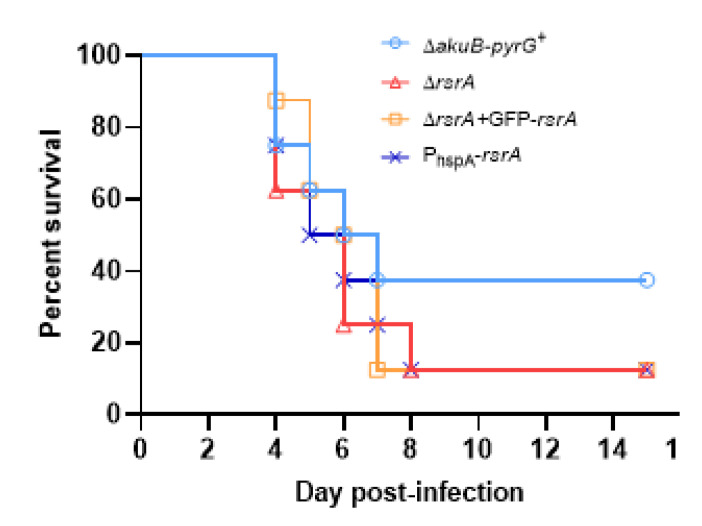
RsrA is not essential for the virulence of *A. fumigatus*. Survival curves of mice (*n* = 8) infected intranasally with 10^6^ conidia of control and *rsrA* mutant strains. The Kaplan–Meier curves were compared using log rank tests in GraphPad Prism v. 8.2.1. No statistical differences were observed between experimental arms.

**Figure 6 jof-06-00285-f006:**
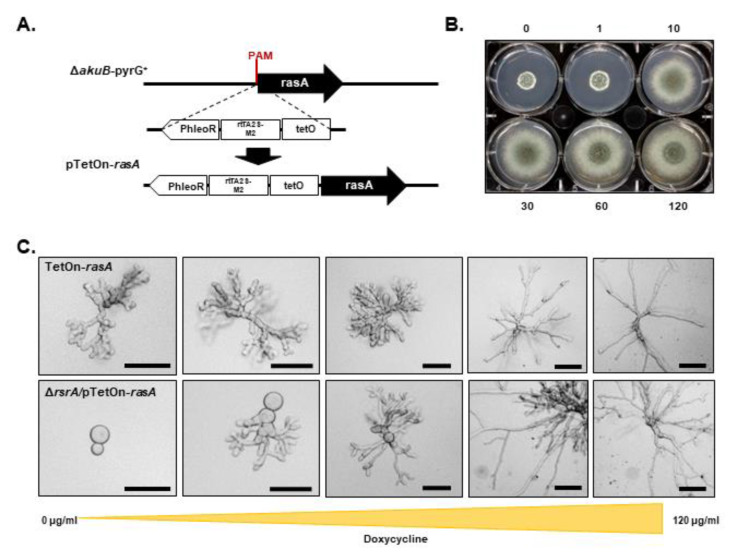
Genetic interaction of RsrA and RasA during polarity establishment in *A. fumigatus*. (**A**) Schematic representation of *rasA* promoter replacement using the tetracycline-inducible promoter (rtTA-M2-TetO) to generate the TetOn-*rasA* or Δ*rsrA*/pTetOn-*rasA* strains. (**B**) Ten thousand conidia from the TetOn-*rasA* strain were point inoculated in the center of GMM agar plates containing 0–120 μg/mL of doxycycline. Photographs were taken after 72 h of growth at 37 °C. (**C**) Two thousand conidia from the TetOn-*rasA* or the double mutant Δ*rsrA*/pTetOn-*rasA* were inoculated in GMM broth with increasing concentrations of doxycycline and incubated at 37 °C for 18 h. In the absence of doxycycline, the TetOn-*rasA* strain (upper panel) is indistinguishable from the *rasA* knockout (Appendix A and Fortwendel et al., 2008), while the double mutant (bottom panel) is unable to establish a polarity axis and produce viable hyphae. Proper hyphal morphologies are recovered at higher concentrations of doxycycline. Scale = 50 μM.

**Figure 7 jof-06-00285-f007:**
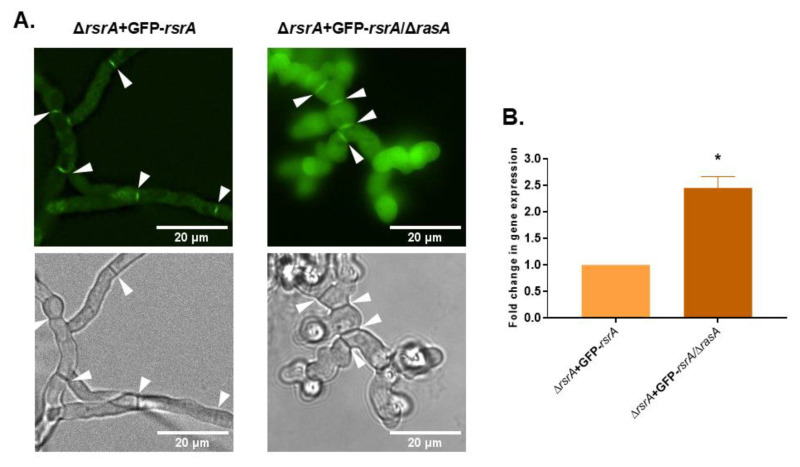
RasA regulates *rsrA* expression, but not its cellular localization. (**A**) One thousand conidia from Δ*rsrA* + GFP-*rsrA* or Δ*rsrA* + GFP-*rsrA*/Δ*rasA* strains were inoculated in liquid GMM and incubated at 37 °C for 16 h. Microphotographs were taken with a Nikon X fluorescence microscope equipped with a GFP filter. (**B**) Deletion of *rasA* causes an increase in *rsrA* transcript levels. Expression of *rsrA* in the Δ*rsrA* + GFP-*rsrA*/Δ*rasA* mutant relative to the parental Δ*rsrA* + GFP-*rsrA* strain was determined by RT-qPCR using *A. fumigatus* β-tubulin (*tubA*) as the endogenous standard. Statistical analysis was performed using unpaired, two-tailed, Student’s *t*-test in GraphPad Prism v. 8.2.1 for Windows (* *p* = 0.0002 in comparison to the parental Δ*rsrA* + GFP-*rsrA* strain). The experiment was performed in technical and biological triplicates. White arrows indicate septa.

**Table 1 jof-06-00285-t001:** Strains used in this study.

Strain	Genetic Background	Source or Reference
KU80Δ*pyrG*	CEA17	Fungal Genetics Stock Center [34]
Δ*akuB-pyrG^+^*	KU80Δ*pyrG*	[11]
Δ*rsrA*	KU80Δ*pyrG*	This study
Δ*rsrA* + GFP-*rsrA*	Δ*rsrA*	This study
*P_hspA_-rsrA*	Δ*akuB-pyrG^+^*	This study
TetOn-*rasA*	Δ*akuB-pyrG^+^*	This study
Δ*rsrA*/pTetOn-*rasA*	Δ*rsrA*	This study
Δ*rasA*	Δ*akuB-pyrG^+^*	This study
Δ*rsrA* + GFP-*rsrA/*Δ*rasA*	Δ*rsrA* + GFP-*rsrA*	This study

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
