# Peer review of "Overexpression of the *Aspergillus fumigatus* Small GTPase, RsrA, Promotes Polarity Establishment during Germination"

_jof, 2020, doi:10.3390/jof6040285_

Round 1
Reviewer 1 Report
The manuscript at hand reports on the identification and genetic characterization of the rsrA gene from the pathogenic fungus Aspergillus fumigatus. In general, the experimental approaches are sound and the findings on the encoded GTPase are of interest to a broad readership. The manuscript is well written and summarizes a fair amount of experimental work. However, in my opinion some of the conclusions are overstated and should be taken with more precaution.
A main concern in this respect is already reflected in the title, where the authors state that polarity establishment during germination is affected at host physiological conditions. This statement is primarily based on the observation, that overexpression of the gene under the control of a heat-shock promoter causes defects at 37°C, but not at 30°C. If I get the conditions right from M&M, overexpression was compared to wild-type at 37°C by RT-PCR. This could be an artefact of the vast overexpression obbserved, which may be much less at 30°C given the promoter employed. As a minimal requirement, I would suggest to at least gather RT-PCR data at the lower temperature. Thus, the observed differences may not be caused by the change in “environmental conditions” or “host physioligcal conditions”, but simply by the level of gene expression. This should be determined.
Moreover, the transcriptional upregulation is likely not accompanied by a similar increase in protein concentration, which is what matters. It would be nice to have some data on the RsrA concentration itself, e.g. by using the GFP fusion in quantitative Western blots. I understand that this would require placing the heat-shock promoter also in front of the fusion construct. While this would be a nice confirmation of the expression data, I will therefore not insist on this experiment, if the authors find it to be too time consuming. More careful phrasing of the conclusions may be sufficient.
A second piece of data I find a bit troublesome is the interpretation of the virulence assays. Although it may be common to use percentages of survival in the medical literature with sample numbers of only 8 mice, each, I find it strongly missleading. It would be much better to speak of “7 out of 8 mice” instead of 87.5%. In this context, I very much doubt the statistical significance of data obtained with such small numbers. So the conclusion from Fig. 5 should be that “there is no drastical difference observed in the virulence of the mutant strain as compared to wild type, as deduced from the survival of 8 mice tested for each strain”. Otherwise, at least 20-30 mice should be tested, which again may not be necessary to make the point.
Minor comments:
Even reading the manuscript at 150% normal size, labels in Figs. 1 and 2 are often too small to be deciphered. Please use larger letter sizes.
In Figs. 1B, 1C and 2D labeling of the axes is redundant. Either it is “Time post inoculation (hours)” or the numbers on the axes are 12h etc..
Line 444/445 “... suggest that RasA regulates RsrA at 444 the level of gene transcription as well as at the protein level for correct subcellular localization”; while I know what the authors want to say, this sounds missleading. The RT-PCR data show that transcription is up-regulated in the rasA mutant, meaning that the wild-type protein negatively regulates transcription. However, there are no experiments indicating that RasA directly regulates the protein level, which would mean regulating the stability of RsrA or being involved in some modification that affects its stability. Thus, RasA affects transcription and presumably this is reflected in the protein level. Here again, it would be nice to use the GFP-tag in a quantitative Western blot to demonstrate that indeed the protein level is affected.
The reference list should be checked carefully. Some species names are not in italics, some journals are not correctly abreviated or spelled out completely.
Author Response
The manuscript at hand reports on the identification and genetic characterization of the rsrA gene from the pathogenic fungus Aspergillus fumigatus. In general, the experimental approaches are sound and the findings on the encoded GTPase are of interest to a broad readership. The manuscript is well written and summarizes a fair amount of experimental work. However, in my opinion some of the conclusions are overstated and should be taken with more precaution.
A main concern in this respect is already reflected in the title, where the authors state that polarity establishment during germination is affected at host physiological conditions. This statement is primarily based on the observation, that overexpression of the gene under the control of a heat-shock promoter causes defects at 37°C, but not at 30°C. If I get the conditions right from M&M, overexpression was compared to wild-type at 37°C by RT-PCR. This could be an artefact of the vast overexpression obbserved, which may be much less at 30°C given the promoter employed. As a minimal requirement, I would suggest to at least gather RT-PCR data at the lower temperature. Thus, the observed differences may not be caused by the change in “environmental conditions” or “host physioligcal conditions”, but simply by the level of gene expression. This should be determined.
A: We understand the reviewer’s concern and thank the reviewer for this important comment. As suggested by the reviewer, we have determined rsrA expression levels at 30°C and we found that rsrA is overexpressed by 22-fold in comparison to the wild type strain. Although this relative expression is lower than that observed at 37°C, these data indicated that the hspA promoter is suitable to drive overexpression even at 30°C. However, as the reviewer suggests, the heightened overexpression that occurs at 37°C (and therefore the polarity establishment phenotype of the pHspA-rsrA mutant) is likely due to the use of a temperature-responsive promoter. We have added this data to Figure 2 and to the reviewed manuscript in lines 294-295. We have also modified the title of our study to reflect this important outcome.
Moreover, the transcriptional upregulation is likely not accompanied by a similar increase in protein concentration, which is what matters. It would be nice to have some data on the RsrA concentration itself, e.g. by using the GFP fusion in quantitative Western blots. I understand that this would require placing the heat-shock promoter also in front of the fusion construct. While this would be a nice confirmation of the expression data, I will therefore not insist on this experiment, if the authors find it to be too time consuming. More careful phrasing of the conclusions may be sufficient.
A: We agree with the reviewer that transcriptional overexpression does not always result in increased protein abundance. However, as part of this study, we have generated multiple strains displaying varying transcriptional expression levels and our phenotype is only observable at the highest levels of gene overexpression. Therefore, it is logical to conclude that, at least at the highest level of overexpression we report here, protein levels are elevated enough to result in the polarity establishment phenotype. That said, we have edited the conclusions and title to be better supported by our findings.
A second piece of data I find a bit troublesome is the interpretation of the virulence assays. Although it may be common to use percentages of survival in the medical literature with sample numbers of only 8 mice, each, I find it strongly missleading. It would be much better to speak of “7 out of 8 mice” instead of 87.5%. In this context, I very much doubt the statistical significance of data obtained with such small numbers. So the conclusion from Fig. 5 should be that “there is no drastical difference observed in the virulence of the mutant strain as compared to wild type, as deduced from the survival of 8 mice tested for each strain”. Otherwise, at least 20-30 mice should be tested, which again may not be necessary to make the point.
A: In the original submission, we stated that differences in virulence were not observed between groups (line 394 of revised manuscript and Figure 5). The numbers of mice utilized are based on published methods of Power Analysis to detect a given difference in virulence between two experimental arms. However, according to the reviewer’s comment, we have now also added the number of mice next to the percentages in lines 394 and 395.
Minor comments:
Even reading the manuscript at 150% normal size, labels in Figs. 1 and 2 are often too small to be deciphered. Please use larger letter sizes.
A: Font size has been increased from Figures 1 and 2.
In Figs. 1B, 1C and 2D labeling of the axes is redundant. Either it is “Time post inoculation (hours)” or the numbers on the axes are 12h etc...
A: Axes labels have been fixed according to the reviewer’s suggestion.
Line 444/445 “... suggest that RasA regulates RsrA at 444 the level of gene transcription as well as at the protein level for correct subcellular localization”; while I know what the authors want to say, this sounds missleading. The RT-PCR data show that transcription is up-regulated in the rasA mutant, meaning that the wild-type protein negatively regulates transcription. However, there are no experiments indicating that RasA directly regulates the protein level, which would mean regulating the stability of RsrA or being involved in some modification that affects its stability. Thus, RasA affects transcription and presumably this is reflected in the protein level. Here again, it would be nice to use the GFP-tag in a quantitative Western blot to demonstrate that indeed the protein level is affected.
A: We understand the limitation of our experiments, and the text has been modified to soften the conclusion (line 466). Indeed, protein quantification would be very useful and it will certainly be a topic of interest in further investigations.
The reference list should be checked carefully. Some species names are not in italics, some journals are not correctly abreviated or spelled out completely.
A: All the references have been checked and corrected accordingly.
Reviewer 2 Report
Dear Authors,
It was a pleasure reading your paper. The data are nicely presented and conclusions are fully supported with results. Unfortunately for the journal readers there is a crucial information missing in the manuscript. Please adress your manuscript with informations about the level of homology between the samll GTPase Rsr1 and Aspergillus fumigatus small GTPase. I think it will be very im portant for readers to see the extent of homology, the overall conservation of the protein domains and proteins structure. Without discussion of this crucial information I am afraid that your paper will not provide the input and the progress beyond state of the art in the field that the readers are seeking.
Best regards
Author Response
It was a pleasure reading your paper. The data are nicely presented and conclusions are fully supported with results. Unfortunately for the journal readers there is a crucial information missing in the manuscript. Please adress your manuscript with informations about the level of homology between the samll GTPase Rsr1 and Aspergillus fumigatus small GTPase. I think it will be very im portant for readers to see the extent of homology, the overall conservation of the protein domains and proteins structure. Without discussion of this crucial information I am afraid that your paper will not provide the input and the progress beyond state of the art in the field that the readers are seeking.
Best regards
A: Information about homology between the Saccharomyces cerevisiae Rsr1 and the A. fumigatus RsrA can be found in lines 209-216 of the original version of the manuscript (lines 215-222 from the revised version) and in Supplemental Figure 1. This figure shows an alignment of both GTPases and the position of the Ras effector domain, the GTP/GDP binding domains and the CAAX box of both proteins.
Round 2
Reviewer 1 Report
The authors have addressed all of my concerns and although they postpone protein quantifications to future studies, I thank the manuscript can be acceptedd in its present form.